# Stable transgenerational epigenetic inheritance requires a DNA methylation-sensing circuit

Ben P. Williams[1] & Mary Gehring [1,2]

Epigenetic states are stably propagated in eukaryotes. In plants, DNA methylation patterns are faithfully inherited over many generations but it is unknown how the dynamic activities of cytosine DNA methyltransferases and 5-methylcytosine DNA glycosylases interact to maintain epigenetic homeostasis. Here we show that a methylation-sensing gene regulatory circuit centered on a 5-methylcytosine DNA glycosylase gene is required for long-term epigenetic fidelity in *Arabidopsis*. Disrupting this circuit causes widespread methylation losses and abnormal phenotypes that progressively worsen over generations. In heterochromatin, these losses are counteracted such that methylation returns to a normal level over four generations. However, thousands of loci in euchromatin progressively lose DNA methylation between generations and remain unmethylated. We conclude that an actively maintained equilibrium between methylation and demethylation activities is required to ensure long-term stable inheritance of epigenetic information.

[1] Whitehead Institute for Biomedical Research, Cambridge, MA 02142, USA. [2] Department of Biology, Massachusetts Institute of Technology, Cambridge, MA 02139, USA. Correspondence and requests for materials should be addressed to M.G. (email: mgehring@wi.mit.edu)

In plants and animals, DNA methylation is important for transcriptional repression of repetitive elements[1,2], silencing of recombination at repetitive regions[3–5], and establishment of genomic imprinting[6]. Disrupted DNA methylation patterns in human cells are associated with multiple cancers[7]. Plant DNA methylation patterns are stably inherited over many generations[8,9], and can persist relatively unchanged in reproductively isolated populations[10]. Symmetric CG methylation is faithfully copied to the daughter strand after DNA replication by the methyltransferase MET1[1], whereas methylation of CHG and CHH sites (where H is any base other than G) is established and maintained by the enzymes CMT3 and CMT2, respectively[11]. All

sequence contexts (CG, CHG, and CHH) can be methylated de novo by the cytosine DNA methyltransferase DRM2, which is recruited to DNA by the RNA-directed DNA methylation pathway (RdDM)[1]. Methylated bases in all sequence contexts can also be actively removed by 5-methylcytosine DNA glycosylases through base excision repair[12]. We recently discovered that the expression of the *Arabidopsis* 5-methylcytosine DNA glycosylase gene *ROS1* is activated by the RdDM pathway and repressed by DNA demethylation by the ROS1 enzyme[13]. These pathways converge at a short 220 bp sequence in the *ROS1* 5′ region[13,14], such that *ROS1* transcriptional output is quantitatively coupled to the methylation level of this sequence. This coupling is conserved in *Arabidopsis lyrata* and may also operate in distantly related maize[13], suggesting it may have an important role in regulating DNA methylation homeostasis. We predicted that methylation-sensitive expression of *ROS1* functions as an epigenetic rheostat for the genome[13].

In this study, we demonstrate that the methylation-sensitive expression of *ROS1*, which couples DNA demethylation activity to the methylation state of the genome, functions to ensure stable epigenetic inheritance. Restoring *ROS1* expression to wild-type levels in a methylation mutant causes abnormal phenotypes that worsen over multiple generations. This also causes widespread methylation losses across the genome. Methylation losses worsen over multiple generations in euchromatic regions, but are progressively reversed in heterochromatin. We conclude that the regulation of *ROS1* by methylation and demethylation pathways functions as a methylation-sensing circuit that buffers against fluctuations or instability in the maintenance of DNA methylation patterns over long periods of time.

## Results

**Abnormal phenotypes in an RdDM mutant that expresses ROS1.** To test the biological significance of the *ROS1* regulatory circuit, we engineered *Arabidopsis rdr2* mutant plants so that the expression of *ROS1* was uncoupled from the methylation status of the genome. *rdr2* mutants do not produce the double stranded RNAs that initiate canonical RdDM, and are characterized by reduced CHH methylation and 10-fold lower *ROS1* expression. We restored *ROS1* expression to wild-type levels in *rdr2* mutants by remethylating the short 220 bp sequence within the endogenous *ROS1* 5′ region (Fig. 1a; Supplementary Fig. 1)[13]. We termed these plants Broken Rheostat (BR) lines because expression of *ROS1* is uncoupled from the genome-wide methylation state. To control for possible *ROS1*-independent effects, we also remethylated the locus in an *rdr2; ros1* double mutant background, which we term BRmut lines (Fig. 1a; Supplementary Fig. 1). We observed that a fraction of BR line plants exhibited an abnormal curled leaf phenotype compared to the parental *rdr2* mutant, which does not display overt morphological differences compared to WT (Fig. 1b, c). Strikingly, both the penetrance and the severity of the curled leaf phenotype increased over multiple generations—even BR lines that initially lacked a phenotype

exhibited a portion of plants with curled leaves in later generations (Fig. 1b, c). By the fifth generation, the majority of individuals from multiple BR lines exhibited curled or severely curled leaves. Conversely, no abnormal phenotypes were apparent in BRmut control lines, indicating that the curled leaf phenotype is dependent on *ROS1* (Fig. 1c).

The triple DNA methyltransferase mutant *drm1; drm2; cmt3 (ddc)* has reduced CHG and CHH methylation and exhibits a curled leaf phenotype similar to what we observed, although there is no variability in the frequency or extent of *ddc* leaf curling[15]. Curled leaves in *ddc* mutants are caused by expression of *SUPPRESSOR OF DDC* (*SDC*), a gene silenced in leaves of wild-type plants by methylation of 5′ tandem repeats by the chromomethyltransferase CMT3 and by RdDM[15]. To determine whether the abnormal leaf morphology in BR lines was correlated with hypomethylation and expression of the *SDC* locus, we performed bisulfite-PCR and sequencing on the *SDC* 5′ region from individual second generation BR line plants that exhibited curled leaves (Supplementary Fig. 2). Consistent with their phenotype, *SDC* CG and CHG methylation was reduced in BR line plants, but not in BRmut line plants, and *SDC* expression was increased between 5- and 100-fold (Supplementary Fig. 2). Because the leaf phenotype of BR lines was variable within and between multiple generations (Fig. 1c), we hypothesized that in this genetic background *SDC* is an unstable epiallele that progressively loses methylation and silencing over multiple generations. To test this hypothesis, RNA and DNA were extracted from a population of pooled plants from second and fifth generation BR lines and used for RT-qPCR and bisulfite-PCR. We observed a significant increase in the abundance of *SDC* transcripts in the fifth generation compared to the second generation for two of three BR lines (Fig. 1d). In addition, the *SDC* locus lost methylation over generational time—the frequency of bisulfite sequencing clones with reduced CG and CHG methylation was higher in the fifth generation compared to the second generation for all three BR lines (Fig. 1e; Supplementary Fig. 2). These changes were not correlated with differences in *ROS1* expression, which remained steady at wild-type levels across generations in the BR lines (Supplementary Fig. 1).

These results suggested two important hypotheses, which we proceeded to test using whole-genome approaches: First, DNA methylation patterns are not stably maintained when *ROS1* expression is uncoupled from the methylation state of the genome and second, the effect of disrupting equilibrium between methylation and demethylation activities cumulatively increases over generational time.

**Uncoupling ROS1 from the methylome causes methylation losses.** We performed whole-genome bisulfite sequencing on individual fourth generation plants of varying phenotypes from four independent BR lines and two BRmut control lines, along with WT, *rdr2*, *ros1*, and *rdr2; ros1* mutants (Supplementary Table 1). A methylation score was computed within overlapping

**Fig. 1** Abnormal phenotypes develop over multiple generations after uncoupling *ROS1* expression from genomic methylation levels. **a** Schematic of the *ROS1* 5′ methylation state and gene expression level in wild-type (WT), *rdr2* mutants, broken rheostat (BR) lines, and control broken rheostat mutant (BRmut) lines. *ROS1* 5′ IR denotes an inverted repeat (IR) transgene complementary to the *ROS1* methylated 5′ region. **b** Representative images of 3-week-old WT (Col-0), *rdr2* and *rdr2;ros1* mutants, as well as second and fifth generation plants from three independent, self-fertilized broken rheostat (BR) lines. **c** Proportion of plants with a curled leaf phenotype, represented by a phenotype score ranging from 0 (no curled leaves) to 5 (all leaves severely curled). Individual plants of median phenotype score for the population were used as parents for the subsequent generation. *N* represents the number of plants examined for each genotype/generation. **d** RT-qPCR quantification of *SDC* transcript levels in RNA from three independent pooled populations of WT, *rdr2*, second, and fifth generation BR line plants. 15–20 plants were combined for each biological replicate. Error bars represent one standard deviation of the mean from three biological replicates. *$p \leq 0.001$ fifth generation compared to second generation (two-tailed *t*-test). **e** Bisulfite-PCR sequencing of the tandem repeats 5′ of *SDC* from DNA extracted from the same pooled tissue used for RT-qPCR as shown in (**d**). Methylation data are shown as the fraction of bisulfite clones that are methylated at different levels. Between 24–37 clones were sequenced for each genotype/generation

300 bp windows to identify differentially methylated regions based on the density of differentially methylated cytosines (see Methods). Overlapping windows were then merged to calculate the total length of DNA with significantly reduced or increased methylation in each sample. All four BR lines showed clear decreases in CG and CHG methylation compared to *rdr2*,

consistent with our findings at *SDC* (Fig. 2a, b). At some loci, CG and non-CG methylation was completely lost (Fig. 2a). In contrast, BRmut lines and *rdr2; ros1* double mutants exhibited slightly higher levels of CG and CHG methylation compared to *rdr2* single mutants (Fig. 2a, b). ROS1 is known to remove methylation from both TEs and gene proximal regions in wild-

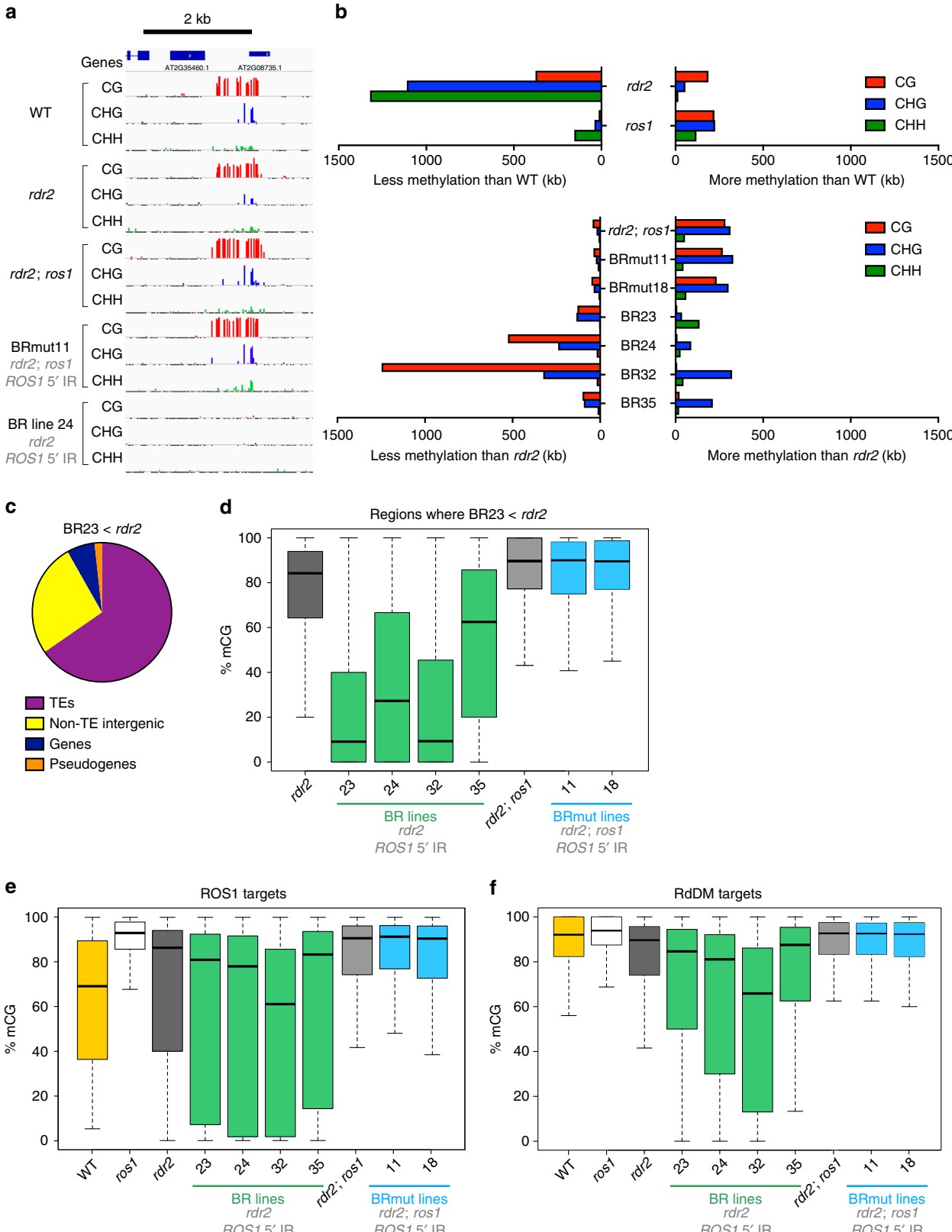

type plants, many of which are also targeted by RdDM[16]. The majority of methylation decreases observed in BR lines were at transposable elements (TEs) and intergenic regions (Fig. 2c; Supplementary Fig. 3). Regions in each individual BR line that had reduced methylation compared to *rdr2* were also hypomethylated in all other BR lines (Fig. 2d; Supplementary Fig. 3), but not in BRmut lines. Thus, independent BR lines lost methylation at shared loci when the methylation-sensing circuit was disrupted. In addition, all four BR lines showed clearly reduced methylation at a subset of ROS1 targets (defined as 1366 loci that significantly gained methylation in any sequence context in *ros1* compared to WT) compared to *rdr2* (Fig. 2e), consistent with restoration of wild-type *ROS1* expression levels in BR lines. There was also a clear reduction in CG methylation at a subset of RdDM targets (6947 loci that are CHH hypomethylated in *rdr2* mutants compared to WT) in all BR lines (Fig. 2f), but not in BRmut lines. Between 13 and 51% of RdDM targets lost more than 20% CG methylation in BR lines. We confirmed these methylation losses at high depth for two loci using bisulfite-PCR (Supplementary Fig. 3). ROS1 is therefore broadly active at a large number of RdDM targets. RdDM mutants like *rdr2* likely retain most CG methylation at these loci because *ROS1* expression is downregulated by the rheostat mechanism. Restoring *ROS1* expression to wild-type levels in *rdr2* upsets the homeostatic balance, tipping it towards demethylation and resulting in widespread methylation losses.

**Disrupting epigenetic homeostasis causes dynamic instability**. As BR lines exhibit unstable and progressively worsening phenotypes due to demethylation of *SDC* (Fig. 1), we sought to determine if methylation is gradually lost over multiple generations elsewhere in the genome. We performed whole-genome bisulfite sequencing for three self-fertilized BR lines on leaves from individual second, third, and fourth generation plants derived by single seed descent. Unexpectedly, at a genome-wide scale, there was the least methylation in second-generation BR plants in all sequence contexts (Fig. 3a; Supplementary Table 1; Supplementary Fig. 4). However, this effect was transient. Average genomic CG and non-CG methylation was partially restored in the third generation, and overall normal levels of methylation were present at most loci in the fourth generation (Fig. 3a; Supplementary Table 1; Supplementary Fig. 4). The reduction of methylation in the second generation occurred at thousands of discrete loci, which were most abundant near the centromeres (Fig. 3b, orange line). By contrast, we observed that the subset of methylation losses shared among all three generations (and thus likely inherited) were abundant across chromosome arms but reduced near the centromere (Fig. 3b, black line). This suggests that although second generation BR lines exhibit the majority of methylation losses around the centromere, losses outside of those regions are more likely to persist heritably in future generations. Consistent with this, chromatin states associated with promoters or with intergenic sequences enriched for the Polycomb mark H3K27me3 showed methylation losses that persisted across four generations (Supplementary Fig. 5), whereas methylation was completely restored over four generations in heterochromatin

(Supplementary Fig. 5), the predominant chromatin state around the centromere (Fig. 3b). To determine whether regions with inherited methylation losses behaved like *SDC* and lost methylation from one generation to the next, we identified regions with reduced methylation in the fourth generation compared to *rdr2* and examined their methylation history in previous generations. Similar to *SDC*, these loci showed a progressive loss of methylation over multiple generations in all three BR lines (Fig. 3c). A subset of genomic loci in chromosome arms therefore behaves in an opposite manner to the majority of the genome, and continues to lose methylation upon disruption of the methylation-sensing circuit.

The methylation dynamics we observed in regions were also present at the level of individual CG pairs (Fig. 3d–f; Supplementary Fig. 6). We performed k-means clustering (with $k = 4$) on individual CGs, comparing their methylation level in the BR lines to the parent *rdr2* genotype (Supplementary Fig. 6). We identified similar clusters in all three BR lines. One cluster (termed Type 1 CGs) behaved like methylcytosines at *SDC*, showing gradually or consistently reduced methylation compared to *rdr2* in all three generations (Fig. 3d). A second cluster (termed Type 2 CGs) behaved like the genomic average, losing methylation in the second generation, but regaining methylation over subsequent generations (Fig. 3d). Type 3 CGs exhibited increased methylation in BR lines compared to *rdr2* (Supplementary Fig. 6). However, the majority of these CGs were also hypermethylated in *rdr2; ros1* mutants and BRmut lines, suggesting they did not change as a result of ROS1 activity. Type 1 CGs were relatively rare around the centromere and were clustered in gene-rich chromosome arms (Fig. 3e; Supplementary Fig. 7), consistent with the location of regions with heritable methylation changes (Fig. 3b). Conversely, Type 2 CGs were abundant in the heterochromatic pericentromere, where CG methylation is most abundant in *rdr2* (Fig. 3e; Supplementary Fig. 7). The methylation restoration at Type 2 CGs, which we validated at high depth for two loci using bisulfite-PCR (Supplementary Fig. 8), occurred in all sequence contexts: CG and CHG (methylated by the enzymes MET1 and CMT3 in WT, respectively) and in CAA and CTA sequences (Supplementary Fig. 4), which are methylated by the enzyme CMT2[17]. Methylation restoration must occur independently of the canonical RdDM pathway because BR lines are *rdr2* mutants. This process is likely mediated by multiple non-RdDM pathways that methylate heterochromatic DNA[18]. Consistent with their targeting by non-RdDM pathways, Type 2 CGs were flanked by cytosines more highly methylated in all sequence contexts in the *rdr2* background than the cytosines flanking Type 1 CGs (Fig. 3f; Supplementary Fig. 7).

## Discussion
We have demonstrated that disrupting the methylation-sensing circuit integrated at *ROS1* results in destabilization of epigenetic states. Our findings suggest that although RdDM pathway mutants, like *rdr2*, have lost substantial CHH methylation, they are in fact epigenetically stable because the *ROS1*-rheostat maintains homeostasis, simply at a lower genome-wide

**Fig. 2** Disrupting the methylation-sensing circuit causes genome-wide methylation losses. **a** A representative genome browser snapshot of whole-genome bisulfite sequencing data from the indicated genotypes. All BR and BRmut line data in this figure are from fourth generation plants. **b** The total sequence length (in kb) that has significantly gained or lost methylation compared to WT (*rdr2* and *ros1* (top)) or compared to *rdr2* (*rdr2; ros1*, BR and BRmut lines (bottom)). Significant methylation changes were determined for 300 bp windows by scoring the number of differentially methylated CG, CHG, or CHH context cytosines as a proportion of total cytosines in each window. **c** Proportion of regions in BR line 23 with reduced methylation compared to *rdr2* (in any sequence context) that overlap different genomic features. **d** Box and whisker plots of methylation levels of individual CGs at the most strongly hypomethylated regions from BR line 23. Whiskers represent 1.5 times the interquartile range. **e** Box and whisker plots of methylation levels for individual CGs overlapping ROS1 targets. **f** Box and whisker plots of methylation levels for individual CGs overlapping RdDM targets

methylation level than in the wild-type. Unstable methylation inheritance has only previously been documented in severe mutants that lose the majority of genomic DNA methylation and have widespread transposition, such as *met1*[19]. Breaking the

rheostat mechanism by restoring *ROS1* expression in *rdr2* creates an unstable state, resulting in abnormal phenotypes that are not observed in *rdr2*. The severity of these phenotypes worsens over generations due to the progressive loss of methylation at *SDC*, a

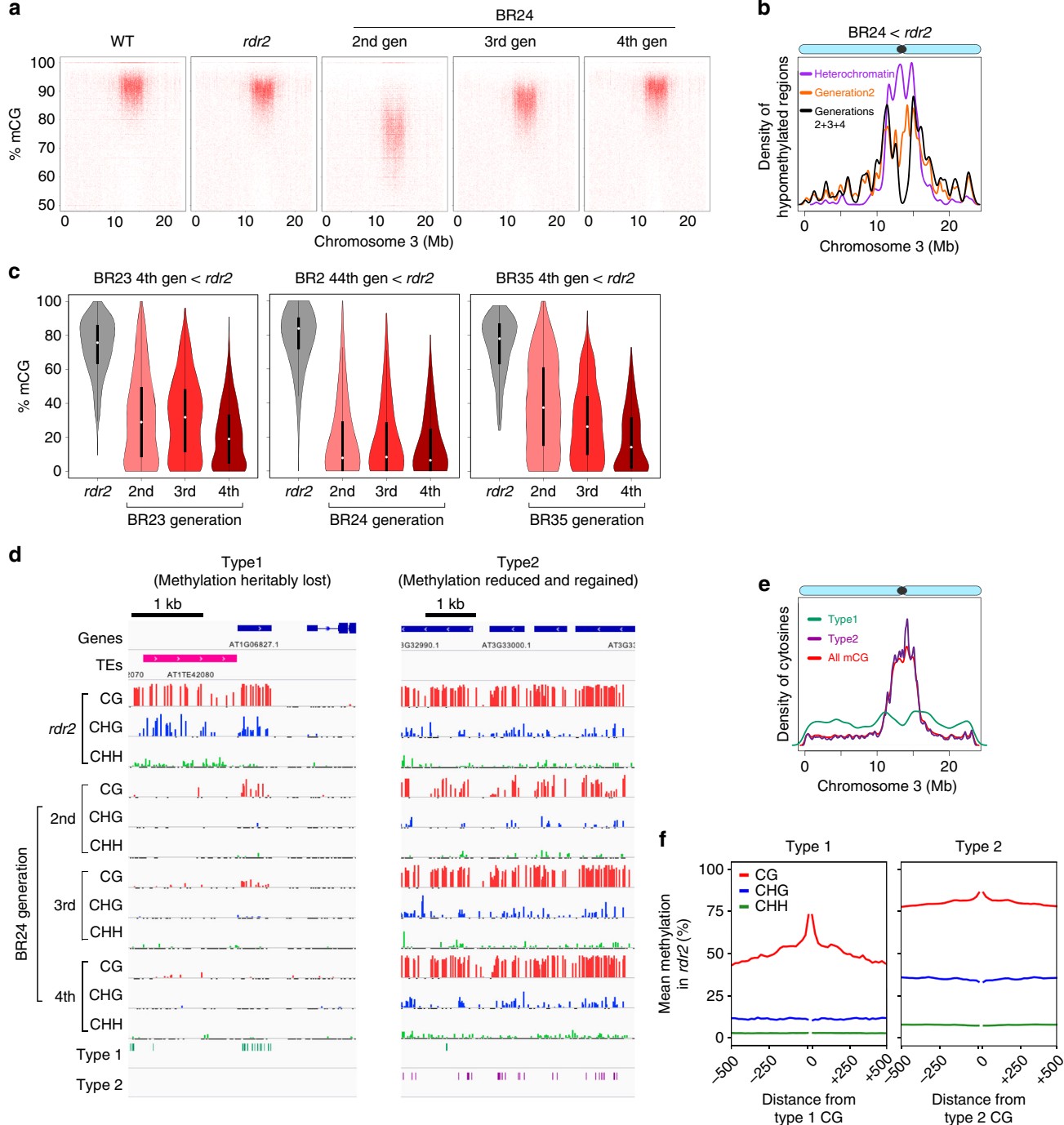

**Fig. 3** Uncoupling methylation and demethylation activities causes dynamic methylome instability over multiple generations. **a** Scatterplots of average CG methylation levels in 300 bp windows across chromosome 3 in wild-type (WT) plants, *rdr2* mutants and three generations of BR line 24. **b** Kernel density plot showing regions that are hypomethylated in BR line 24 compared to *rdr2* on chromosome 3. The orange line denotes all regions hypomethylated in the second generation, whereas the black line shows the subset of regions that remain hypomethylated in the third and fourth generations. The purple line denotes the density of heterochromatin across chromosome 3. **c** Violin plots of average CG methylation levels at loci hypomethylated in fourth generation BR plants compared to *rdr2*. **d** Representative snapshots of two DNA methylation dynamics that occur over multiple generations when *ROS1* expression is restored in *rdr2* mutants. Type 1 CGs exhibit gradual or immediate losses of methylation that are inherited over multiple generations. Type 2 CGs are hypomethylated in the second generation but regain methylation over multiple generations. **e** Kernel density plots showing the distribution of Type 1 and Type 2 CGs from BR line 24 over chromosome 3. The red line denotes the density of all methylated CGs in *rdr2*. **f** Average methylation levels of the *rdr2* mutant in sequences proximal to Type 1 and Type 2 CGs. Methylation levels are calculated for sequences up to 500 bp upstream and downstream

phenomenon that is evident at many other loci (Fig. 3c). Yet, mechanisms exist to counteract unstable inheritance (Fig. 3a), restricting the degradation of methylation to the subset of the genome that is primarily methylated solely by the RdDM pathway in the wild-type[20,21]. Progressive remethylation of hypomethylated DNA over a few generations has been observed previously, but was proposed to be mediated by RdDM and did not occur when *RDR2* was mutated[22–25]. Our results demonstrate that the canonical RdDM pathway is in fact dispensable for the reversion of methylation losses in heterochromatin. This implies that other methyltransferases, perhaps CMT2, which is primarily active in heterochromatin[11,18], are capable of reversing reduced methylation. Despite the operation of this apparent push-back mechanism, the methylation landscape does not return to the parental *rdr2* state, particularly in gene-rich regions, where methylation patterning becomes increasingly corrupted over time and results in DNA methylation losses that persist as long-term "epigenetic scars". Thus, a tightly coupled equilibrium between demethylation and methylation pathways is required to preserve the transgenerational stability of the epigenome. It is likely that similar safeguards exist to preserve epigenetic inheritance over multiple cell divisions and generations in other eukaryotes.

## Methods

**Plant material**. All plants were grown in a Conviron BDR16 growth chamber with 16 h of light (120 µMol), 22 °C, and 50% humidity. The *rdr2* and *ros1* mutant alleles used in this study were *rdr2-1* and *ros1-7* as described[13] and are in the Col-0 background. *ros1-7* has a missense mutation in the DNA glycosylase catalytic domain (E956K). Wild-type and homozygous mutant plants used in this study were all derived from the self-fertilized progeny of an *rdr2; ros1* heterozygous parent. The *ROS1* 5′ inverted repeat (IR) transgene used to generate BR lines and BRmut lines was previously described[13]. BR lines were generated by transforming *rdr2* mutants with the IR transgene by floral dipping, whereas BRmut lines were generated by floral dipping *rdr2; ros1* mutants. T₁ generation transgenic lines were selected on 0.5 × MS with 25 µg/ml kanamycin and 25 µg/ml hygromycin. T₂ transgenic lines were initially screened for single insertions by antibiotic selection. T₂ plants used in subsequent assays were grown directly on soil and the presence of the IR transgene was determined by PCR. Each BR or BRmut line is derived from an independent remethylation of *ROS1* and represents biological replicates.

**Plant phenotyping**. Three-week-old adult plants were imaged using a Nikon 1 J1 camera under ambient lighting. A phenotype score (ranging from 0 to 5) was designated to quantify the severity of curled leaf phenotypes in transgenic lines. All plant genotype names were blinded by an independent researcher to ensure unbiased assessment of phenotypes. Plants were manually assessed and scored according to the following criteria: 0 = no leaves show curled laminae; 1 = one or more leaves show curled laminae; 2 = more than half of leaves show some degree of curled laminae; 3 = more than half of leaves are curled and at least one leaf has severely curled laminae; 4 = More than half of leaves have severely curled laminae; 5 = all leaves have severely curled laminae.

Severely curled refers to leaves in which the adaxial surface has curled sufficiently to touch the abaxial surface (e.g., curvature of almost 360 degrees).

**Locus-specific bisulfite-PCR and sequencing**. DNA was extracted from 3-week-old leaves of individual BR line and BRmut line plants, as well as WT (Col-0), *rdr2* and *rdr2; ros1*. 200 ng of DNA was bisulfite converted using an Invitrogen MethylCode bisulfite conversion kit. PCR was performed using 2 µl bisulfite converted DNA and ExTaq HS DNA polymerase (Clontech). Primers are listed in Supplementary Table 2. PCR cycling conditions were 95 °C for 3 min, 40 cycles of (95 °C for 15 s, 50 °C for 30 s, 67.5 °C for 30 s, 72 °C for 30 s), followed by 67.5 °C for 10 min. PCR products were gel extracted and cloned using a CloneJET PCR cloning kit (Thermo Scientific). Between 24 and 37 individual clones for each sample were amplified by colony PCR and sequenced using a T7 promoter primer. Sequenced clones were aligned using MUSCLE[26] and methylation of cytosines was calculated using Cymate[27]. For the analysis of *SDC* methylation in second (T₂) and fifth (T₅) generation BR lines, DNA was extracted from 2-week-old leaf tissue pooled from approximately 20 plants of each genotype/generation. Bisulfite conversion and PCR were performed as specified above. Pooled tissue was divided and the same tissue was used for the extraction of RNA to perform RT-qPCR, as specified below.

**RT-qPCR**. For the data in Supplementary Figs. 1 and 2, RNA was extracted from 3-week-old leaves of individual T₂ BR line and BRmut line plants, as well as WT, *rdr2* and *rdr2; ros1* mutants. Genomic DNA was removed using amplification-grade

DNAseI (Invitrogen). cDNA was synthesized from 1 µg RNA using Superscript II reverse transcriptase (Invitrogen) according to manufacturers' instructions, selecting for polyadenylated transcripts using an oligo-dT primer. qPCR was performed using Fast SYBR-Green PCR master mix (Applied Biosystems) and a StepOne Plus Real-Time PCR system (Applied Biosystems). Primers were designed as previously specified[13]. The reference gene AT1G58050 was used to normalize all reactions, as described[28] (Supplementary Table 2). All reactions were performed in technical triplicate, except for the data in Fig. 1d, which were performed in biological triplicates. For the quantification of *SDC* transcripts in T₂ and T₅ generation BR lines, RNA was extracted from pooled tissue from leaves of ~20 2-week-old plants of each genotype/generation. Three pooled samples were collected for each genotype to provide biological replicates.

**Whole-genome bisulfite sequencing**. Single generation samples (Fig. 2): DNA was extracted from 3-week-old whole rosettes of three independent T₄ BR lines, two independent T₄ BRmut lines, *ros1*, *rdr2*, and *rdr2; ros1* mutants and WT Col-0. Overall, 200 ng DNA was bisulfite converted using an Invitrogen MethylCode bisulfite conversion kit. Libraries were constructed using an Illumina TruSeq DNA methylation kit and sequenced on an Illumina HiSeq2000 using a 40 × 40, 50 × 50, 75 × 75 or 100 × 100 bp paired-end protocol at the Whitehead Institute Genome Technology Core (see Supplementary Table 1 for details). Read quality was assessed with FastQC and adapters were trimmed using Trim Galore (Babraham Bioinformatics). Reads were mapped to the TAIR10 genome using Bismark[29], allowing for 1 mismatch for 40 bp and 50 bp reads and 2 mismatches for 75 and 100 bp reads. Bismark was also used to remove PCR duplicates, and the Bismark methylation extractor function was used to obtain methylation data from all mapped reads. Efficiency of bisulfite conversion was verified by quantifying the percentage of methylation mapped to the chloroplast (Supplementary Table 1). Multiple generation samples (Fig. 3): DNA was extracted from the fifth and sixth true leaves of individual T₂, T₃, and T₄ plants from three BR lines. T₂, T₃, and T₄ generation plants were derived from a single seed descent experiment (for example, the T₃ plants were the self-fertilized progeny of the sequenced T₂ plants). Bisulfite libraries were constructed and mapped as described above.

**Identifying differentially methylated regions**. To identify regions of the genome that were differentially methylated between two samples, a "window score" was calculated based on the number and density of differentially methylated cytosines within a given window. This approach was devised to minimize the false-positive discovery of differentially methylated regions in which only a small number of cytosines are differentially methylated. The following conditions were imposed: First, the genome was divided into 300 bp tiled windows that overlapped by 150 bp. Second, methylation data for cytosines in the CG context were averaged with the symmetric cytosine on the opposite strand. This resulted in one methylation value for each CG pair, so that cytosines in a symmetric CG dimer are not treated as independent. Third, only cytosines covered by 5 or more reads in both samples were considered. Fourth, cytosines were considered differentially methylated if their percentage methylation level differed by the following intervals: CG pairs: >30%, CHG: >25%, CHH: >20% (window scores are calculated separately for each methylation context). A methylation score was calculated by assigning +1 for every hypermethylated cytosine or CG pair and −1 for every hypomethylated cytosine or CG pair in a given window. This score was then multiplied by the proportion of cytosines in the window (with sufficient data for analysis) that were differentially methylated. The higher the score above 0, the greater the confidence a window is hypermethylated. The lower the score below 0, the greater the confidence a window is hypomethylated. This approach has several advantages over many conventionally used methods. First, two cytosines in a symmetric CG dimer are not treated as independent data points. Second, cytosines that are not well covered in both samples do not affect the window score. In the analysis presented in Figs. 2 and 3, hypermethylated regions were chosen as windows with a score of 3 or greater, and hypomethylated regions were chosen as windows with a score of −3 or lower. Bedtools intersect was used to determine the proportion of differentially methylated windows overlapping TEs, genes, non-TE intergenic regions and pseudogenes according to the Araport 11 genome annotation. To quantify the methylation state of individual CGs at ROS1 and RdDM targets (Fig. 2e, f), a set of ROS1 and RdDM target loci was defined based on their window score. ROS1 targets were defined as merged windows with a score of 3 or greater in any cytosine context in *ros1* relative to WT (1366 loci identified). RdDM targets were defined as merged windows with a score of −3 or lower in the CHH context in *rdr2* compared to WT (6947 loci identified). Boxplots in Fig. 2e, f show the percentage average methylation value of all individual CGs covered by 5 or more reads that are located within ROS1 or RdDM targets.

***k*-means clustering**. Clustering was performed on symmetric CG pairs. For each BR line, only CGs covered by 5 or more reads in *rdr2* and in all three generations were considered (approximately one quarter of all possible cytosines met this coverage threshold). Methylation values for each BR line sample were subtracted from *rdr2* and the values of methylation differences were clustered by a *k*-means algorithm using Cluster 3.0 (Stanford University). A total of 1000 iterations were run and the iteration that produced the smallest within-cluster distances was

chosen. Clustering was performed with different values for *k* (4,5,6,8). Results obtained with *k* = 4 were chosen, as increasing *k* above 4 did not identify new variants. Methylation differences at the cytosines identified in each cluster were calculated between *rdr2* and *rdr2; ros1*, and between *rdr2* and BRmut lines, to control for changes to methylation independent of *ROS1*. The distance between the closest Type 1 or Type 2 cytosines was calculated using bedtools closest. The distance distributions expected by chance were calculated from five randomly selected datasets (of the same size as the Type 1 or Type 2 dataset) of cytosines methylated in *rdr2*. Kernel density plots of cytosines from different clusters were generated using R. Bedtools intersect was used to determine the proportion of cytosines in each cluster overlapping TEs, genes, non-TE intergenic regions and pseudogenes according to the Araport 11 genome annotation. The methylation profile of sequences surrounding Type 1 or Type 2 cytosines was generated as described[30].

**Other analyses of bisulfite-seq data.** The methylation state of different chromatin states was determined using the chromatin states identified by Sequeira-Mendes et al.[31] Chromatin states 1 and 2 were combined to represent "Transcription start sites and promoters", chromatin states 4 and 5 were combined to represent "Intergenic H3K27me3" and chromatin states 8 and 9 were combined to represent "heterochromatin". Violin plots of the percent methylation of individual CGs (covered by 5 or more reads) were generated using R. The percentage methylation of different sequence contexts over whole chromosomes (Supplementary Fig. 4) was calculated in R using a moving average function, averaging methylation over 10 kb windows overlapping by 5 kb.

**Data availability.** All sequencing data is available in NCBI GEO under record GSE104240. The authors declare that all other data supporting the findings of this study are available within the manuscript and its supplementary files or are available from the corresponding author upon request.

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

## Acknowledgements

We thank Katherine Novitzky and Elena Kingston for valuable assistance with locus-specific bisulfite-PCR and Colette L. Picard for advice on bioinformatics analysis. Research reported in this publication was supported by the National Institute of General Medical Sciences of the National Institutes of Health under award R01GM112851 to M. G.

## Author contributions

B.P.W. and M.G. conceived and designed experiments, B.P.W. performed experiments and analyzed data; B.P.W. and M.G. wrote the paper.

## Additional information

**Competing interests:** The authors declare that they have no competing financial interests.

