## [Peer Review File · Nature Communications]

Reviewers' comments:

Reviewer #1 (Remarks to the Author):

The manuscript by Williams and Gehring, „Stable transgenerational epigenetic inheritance requires a DNA methylation-sensing circuit“, describes how interfering with expression of gene encoding a DNA demethylase interrupts an otherwise functional feedback regulatory mechanism that maintains DNA methylation between generations. The authors show that different regions in the genome differ in the degree and stability of the methylation changes controlled by this mechanism.

This is a nice and interesting continuation of previous work by the authors (Williams et al. 2015) and a complementation of data obtained with a mutant lacking a DNA methylation maintenance enzyme (e.g. Teixeira et al. 2009). These new data provide solid evidence for an important role of one of the glycosylases, itself transcriptionally controlled by methylation at the promoter region of its gene, for the balance of installation and removal of cytosine methylation to achieve faithful transmission of epigenetic information to subsequent generations. The data also indicate an important role for the non-RdDM pathway of (re-)methylation.

The manuscript is clearly written, the experiments are well designed and documented. In addition to sufficient sample numbers, good controls and adequate statistical procedures, I like the high standard of phenotype analysis with blinded evaluation and clear criteria, as well as the clear and controlled pedigree of the plants analyzed. The conclusions are supported by the data. Although DNA demethylation is mechanistically different in other organisms, this is another persuasive example for the power of genetic and molecular approaches possible in plants to expand our understanding of epigenetic inheritance.

I have minor suggestions to improve the manuscript:

The pleonasm in the title should be removed: inheritance is per definition transgenerational; stable epigenetic inheritance is sufficient in this context.

The nature of the “control lines” is well explained and illustrated in Supplementary Figure 1 but not so clear in the main text and the figures. What about calling them BRmut lines throughout, better reflecting that the ROS is expressed but does not produce functional protein?

p.4 l.2: make clear that the SDC gene is silenced in leaves OF WILDTYPE PLANTS by methylation...

Figure 2 b: try to explain the procedure better, it took me a long time to understand this graph.

It is difficult for the reader to understand the choice of the line and the generation between different types of analysis. This seems a bit arbitrary. Just one example: Supplementary Figure 2 correlates SDC methylation with SDC expression, but shows control 3 and BR 23#3 only in panel s, and for lines BR 24 and 35 the second generation in panel b and the third generation in panel c. I am not asking for new experiments but rather for a reduction on complete and congruent data sets. Less is more, also here.

Please explain the choice of the locus for the browser snapshot in Figure 2a and add the generation of the material for the BS-seq (fourth) to the legend.

Please describe the normalization procedure for the RT-qPCR of gene expression analyses in the respective method section.

Reviewer #2 (Remarks to the Author):

Expression of the DNA glycosylase ROS1 in Arabidopsis is regulated by the RNA-dependent DNA methylation (RdDM) pathway, where RdDM is required for ROS1 activation, while ROS1 is negatively regulating its own expression. The authors of this manuscript investigated the consequences of disrupting this circuit by expressing ROS1 in an *rdr2* mutant background (deficient in the production of double-stranded RNAs, the precursors of small RNAs), but inducing DNA methylation by expressing an artificial double stranded RNA targeting the promoter of ROS1. Expressing this construct causes phenotypes that progressively become stronger over generations, which the authors associate with increased DNA demethylation at euchromatic regions and expression of SDC that previously has been associated with some of the observed phenotypes. The authors interpret their findings that an actively maintained equilibrium between methylation and demethylation is required to ensure stable inheritance of DNA methylation. My main problem with this manuscript is that I fail to see the major novelty in this finding, expressing a DNA glycosylase targeting methylated regions is expected to cause methylation loss if the RdDM pathway is defective.

Other comments:

1. The authors did not provide evidence that expression of ROS1 does not change over generations, which would provide a simple explanation for the observed worsening of phenotypes over generations.
2. Page 4, Figure 1d: I am not convinced about the claimed increase in SDC expression between first and fifth generation. In Suppl. Figure 2c it is obvious that SDC expression differs between individuals of the same line; since the authors pooled plants to generate data of Fig. 1d, the observed differences could be a consequence of expression changes between individuals rather than a generation consequence. Apparently the authors mixed homo- and heterozygous individuals, which likely contributed to the observed variability. Furthermore, the data shown are technical, not biological replicates, limiting the value of the conclusions.
3. Figure 1e: The figure would be more conclusive if the authors represent the data as % methylation, instead of showing the fractions of clones having a certain methylation status.
4. Fig. 2d, e, f: The statement that regions with reduced methylation in each BR line were also significantly hypomethylated in all other BR lines is not very obvious based on Fig. 2d and Suppl. 3. It would be more convincing to define how many regions show a significant loss of DNA methylation in all lines and show the overlap of those regions. It is neither obvious that all lines lost methylation at ROS1 and RdDM target regions. The difference is clear for line 32, but not for the other lines.

Minor point:

Page 2: "This coupling is conserved among multiple flowering plant species" is an overstatement; so far clear evidence for coupling has been shown for *A. lyrata* and suggested for maize.

Response to Reviewers' Comments

Williams and Gehring

October 3, 2017

We thank the reviewers for their constructive comments and criticisms. Addressing these has improved the manuscript.

Reviewer #1 (Remarks to the Author):

The manuscript by Williams and Gehring, "Stable transgenerational epigenetic inheritance requires a DNA methylation-sensing circuit", describes how interfering with expression of gene encoding a DNA demethylase interrupts an otherwise functional feedback regulatory mechanism that maintains DNA methylation between generations. The authors show that different regions in the genome differ in the degree and stability of the methylation changes controlled by this mechanism.

This is a nice and interesting continuation of previous work by the authors (Williams et al. 2015) and a complementation of data obtained with a mutant lacking a DNA methylation maintenance enzyme (e.g. Teixeira et al. 2009). These new data provide solid evidence for an important role of one of the glycosylases, itself transcriptionally controlled by methylation at the promoter region of its gene, for the balance of installation and removal of cytosine methylation to achieve faithful transmission of epigenetic information to subsequent generations. The data also indicate an important role for the non-RdDM pathway of (re-)methylation.

The manuscript is clearly written, the experiments are well designed and documented. In addition to sufficient sample numbers, good controls and adequate statistical procedures, I like the high standard of phenotype analysis with blinded evaluation and clear criteria, as well as the clear and controlled pedigree of the plants analyzed. The conclusions are supported by the data. Although DNA demethylation is mechanistically different in other organisms, this is another persuasive example for the power of genetic and molecular approaches possible in plants to expand our understanding of epigenetic inheritance.

Response: Thank you for the positive feedback.

I have minor suggestions to improve the manuscript:

The pleonasm in the title should be removed: inheritance is per definition transgenerational; stable epigenetic inheritance is sufficient in this context.

Response: We agree that the title has a somewhat redundant use of words. However, many papers in the field use the term 'epigenetic inheritance' to refer to inheritance over mitotic cell divisions. We therefore decided to include the word transgenerational to clarify that our study reports on inheritance over generations.

The nature of the "control lines" is well explained and illustrated in Supplementary Figure 1 but not so clear in the main text and the figures. What about calling them BRmut lines throughout, better reflecting that the ROS is expressed but does not produce functional protein?

Response: We thank the reviewer for this suggestion. We have changed the name of the control lines to BRmut lines throughout the text and figures.

p.4 l.2: make clear that the SDC gene is silenced in leaves OF WILDTYPE PLANTS by methylation...

Response: This has been corrected.

Figure 2 b: try to explain the procedure better, it took me a long time to understand this graph.

Response: We have included the following sentences in the main text (p. 6) to explain the approach taken in more detail:

A methylation score was computed within overlapping 300 bp windows to identify differentially methylated regions based on the density of differentially methylated cytosines (see Methods). Overlapping windows were then merged to calculate the total length of DNA with significantly reduced or increased methylation in each sample.

It is difficult for the reader to understand the choice of the line and the generation between different types of analysis. This seems a bit arbitrary. Just one example: Supplementary Figure 2 correlates SDC methylation with SDC expression, but shows control 3 and BR 23#3 only in panel s, and for lines BR 24 and 35 the second generation in panel b and the third generation in panel c. I am not asking for new experiments but rather for a reduction on complete and congruent data sets. Less is more, also here.

Response: The data in Supplementary Figure 2 is showing individual plants from the second generation, rather than a mixture of plants from multiple generations. We apologize that this was not clearer and we have amended the figure to annotate it more accurately. We have also removed data from BRmut lines not present in both graphs to reduce the number of data points, as suggested.

Please explain the choice of the locus for the browser snapshot in Figure 2a and add the generation of the material for the BS-seq (fourth) to the legend.

Response: We attempted to choose a region that was, a) clearly representative of the most common methylation changes we observe and b) easily visible to the reader. We observed methylation losses in intergenic euchromatin frequently in all BR line samples (see Figure 2c), thus we think this snapshot is broadly representative of the genome-wide changes. We have added the word 'representative' to the legend to make this clearer. We have also annotated the samples as fourth generation in the legend.

Please describe the normalization procedure for the RT-qPCR of gene expression analyses in the respective method section.

Response: To normalize our RT-qPCR we used a reference gene (AT1G58050) that was experimentally validated by Czechowski *et al* (Plant Physiology 2005) to have very consistent, moderate transcript abundance across all tissues in Arabidopsis. We have now included the details of this in the methods section.

Reviewer #2 (Remarks to the Author):

Expression of the DNA glycosylase ROS1 in Arabidopsis is regulated by the RNA-dependent DNA methylation (RdDM) pathway, where RdDM is required for ROS1 activation, while ROS1 is negatively regulating its own expression. The authors of this manuscript investigated the consequences of disrupting this circuit by expressing ROS1 in an *rdr2* mutant background (deficient in the production of double-stranded RNAs, the precursors of small RNAs), but inducing DNA methylation by expressing an artificial double stranded RNA targeting the promoter of ROS1. Expressing this construct causes phenotypes that progressively become stronger over generations, which the authors associate with increased DNA demethylation at euchromatic regions and expression of SDC that previously has been associated with some of the observed phenotypes. The authors interpret their findings that an actively maintained equilibrium between methylation and demethylation is required to ensure stable

inheritance of DNA methylation. My main problem with this manuscript is that I fail to see the major novelty in this finding, expressing a DNA glycosylase targeting methylated regions is expected to cause methylation loss if the RdDM pathway is defective.

Response: There are two considerably novel findings in our paper that extend knowledge beyond the above summary. RdDM mutants are hypomethylated and 5-methylcytosine DNA glycosylase mutants are hypermethylated, but the epigenomes of these mutants represent stable steady-states, in which genomic methylation is unchanging and accurately inherited over generations. In addition, phenotypic changes in these mutants are minimal and consistent over generations. Our study is novel in that we show that restoring expression of *ROS1* in *rdr2* mutants creates an unstable state. Both genomic methylation patterns and phenotypes dynamically change and worsen every generation. We think this is an important finding because it indicates a role for the tight regulation of *ROS1* expression in maintaining faithful DNA methylation inheritance. Previous studies have described mechanisms to explain how existing methylation is copied at DNA replication, but the way in which *de novo* methylation and demethylation pathways combine to create a stably inherited genome were previously unknown. Unstable methylation inheritance has only previously been documented in severe mutants that lose the majority of methylation and have widespread transposition, such as *met1* and *ddm1*. Our study shows that the interplay between RdDM and *ROS1* (which are thought to affect a subset of the methylation maintained by *MET1*) is surprisingly also essential for proper inheritance of epigenetic information. The expression of *ROS1* is determined by a simple mechanism with positive and negative input from methylation and demethylation. Our study shows that this regulatory circuit is important for long-term epigenetic inheritance. We think this system buffers against fluctuations or instability in the maintenance of methylation patterns.

A second novel aspect of our study is the way in which the genome responds in heterochromatin to counteract this instability. Previous studies have implicated the RdDM pathway as important for restoring hypomethylated DNA to a methylated state in heterochromatic regions. However, our study shows that this response can be independent of *pol4/rdr2*-RdDM activity. This also highlights an interesting aspect of plant epigenetics, which is that methylation patterns in gene-rich regions appear to be more vulnerable to changes in the balance between *ROS1* and RdDM, whereas heterochromatin is only perturbed by such an imbalance in the short-term.

Other comments:

1. The authors did not provide evidence that expression of *ROS1* does not change over generations, which would provide a simple explanation for the observed worsening of phenotypes over generations.

Response: We conducted a qRT-PCR experiment on second, third and fourth generation plants from 3 independent BR lines. We found that the expression of *ROS1* was very similar to wild-type in all plants, and was consistent and unchanging over all three generations. We therefore conclude that changes in *ROS1* expression do not contribute to the observed methylation changes in BR lines. We have included these data in Supplementary Figure 1d.

2. Page 4, Figure 1d: I am not convinced about the claimed increase in SDC expression between first and fifth generation. In Suppl. Figure 2c it is obvious that SDC expression differs between individuals of the same line; since the authors pooled plants to generate data of Fig. 1d, the observed differences could be a consequence of expression changes between individuals rather than a generation consequence. Apparently the authors mixed homo- and heterozygous individuals, which likely contributed to the observed variability. Furthermore, the data shown are technical, not biological replicates, limiting the value of the conclusions.

Response: The reviewer is correct that *SDC* expression differs between individuals of the same line in the same generation – this was the reason we chose to pool plants in each generation to determine *SDC* expression at the population, rather than individual, level. Selecting individual plants for expression analysis could result in a biased outcome (e.g. if we selected a plant with a very strong curled leaf phenotype it would have high *SDC* expression, but this would not necessarily be representative of that generation as a whole). Assaying the generation on the population scale by pooling ~20 plants is the only way to control for individual variation. We also note that the data in Figure 1d are plotted on a log₁₀ scale, and so the large differences between generations 2 and 5 (up to 5-fold) were unlikely to be due single plants affecting the total transcript abundance of the pool.

While the second-generation plants would have indeed been a mix of heterozygotes and homozygotes for the transgene that induces *ROS1* methylation, we are confident that this did not affect the expression of *ROS1* between plants. We have measured the expression and methylation of *ROS1* in heterozygous and homozygous individuals several times, and they always have comparable abundance of *ROS1* transcripts (see Supplementary Figure 1b and 1d, for example). We therefore think that the transgene does not have an additive effect on rescuing *ROS1* expression when homozygous.

However, given the concerns expressed with these data, we repeated the qRT-PCR experiment with 2 more independent sets of pooled samples, totaling 3 biological replicates for each of the three BR lines (which are themselves biological replicates, as they represent independent transformations of the *ROS1* 5' IR transgene). Figure 1d now shows the mean and standard deviation between biological replicates, and supports the same conclusion as the prior data.

3. Figure 1e: The figure would be more conclusive if the authors represent the data as % methylation, instead of showing the fractions of clones having a certain methylation status.

Response: We have calculated the percentage methylation for each pooled sample as suggested and this information is now included in Supplementary Figure 2d. However, we would like to expand upon our rationale for including the “fraction of clones” plot as part of Figure 1, instead of a plot of percentages:

Similar to the data shown in Figure 1d, we were concerned that selecting individuals could result in a biased outcome because there is variation between the phenotypes of each individual within a generation. The bisulfite PCR data in Fig. 1e is therefore also from pooled tissue of ~20 plants (the same tissue used for one of replicates of the RT-qPCR in Fig. 1d). Because each bisulfite PCR clone comes from an individual cell and we sampled equal amounts of tissue from each individual, we expect that on average, each clone represents one plant in the pool (we sequenced twice as many clones as individuals in the pool, doubling the probability that each plant is represented).

Our “fraction of clones” method is therefore a method of representing the variation within the population, and the frequency of demethylated plants in each pool. While this is a more complex graph than an averaged percentage across all clones, we believe it presents information more relevant to the experiment conducted. We also think that it complements the data shown in Figure 1c, which shows the structure of phenotypic variation in the population.

4. Fig. 2d, e, f: The statement that regions with reduced methylation in each BR line were also significantly hypomethylated in all other BR lines is not very obvious based on Fig. 2d and Suppl. 3. It would be more convincing to define how many regions show a significant loss of DNA methylation in all lines and show the overlap of those regions. It is neither obvious that all lines lost methylation at *ROS1* and *RdDM* target regions. The difference is clear for line 32, but not for the other lines.

Response: Overlap statistics for more than 2 samples are problematic, as a slight reduction in coverage may disqualify one sample from significance, even if that region is hypomethylated in all other samples. This approach usually leads to a considerable underestimation of the similarity between samples. Furthermore, the definition of significance requires binary cutoffs to be assigned (a decision with inherent bias), despite fully quantitative data being available. We therefore opted to present the distribution of methylation levels for each line as a boxplot (Fig. 2d and Supplementary Fig. 3), to show readers the effect size and difference in methylation of each line compared to *rdr2*.

In Figure 2d and Supp. Fig 3, all BR lines show significantly reduced methylation compared to *rdr2* ($p = <1 \times 10^{-14}$, Mann-Whitney U test with Bonferroni correction). However, we do not believe p-values are particularly instructive for such large sample sizes, so we did not include these in the manuscript.

Regarding the loss of methylation at ROS1 and RdDM target regions, All BR lines show statistically reduced methylation compared to *rdr2* with $p = <1 \times 10^{-16}$. The lower 25th percentile values in Figure 2E and F show that methylation has been lost at a large subset of ROS1 and RdDM targets in each BR line. We recognize that this was minimally discussed in the main text, so we have expanded this section to make it more clear that BR lines lose methylation at a subset of these regions, and also stated the percentage of RdDM targets that significantly lose methylation in BR lines. We copy the revised text below:

Additionally, all four BR lines showed clearly reduced methylation at a subset of ROS1 targets (defined as 1366 loci that significantly gained methylation in any sequence context in *ros1* compared to the WT) compared to *rdr2* (Fig. 2E), consistent with restoration of wild-type *ROS1* expression levels in BR lines. There was also a clear reduction in CG methylation levels at a subset of all RdDM targets (6947 loci that are CHH hypomethylated in *rdr2* mutants) in all BR lines (Fig. 2F), but not in BRmut lines. Between 13-51% of RdDM targets lost more than 20% CG methylation in BR lines.

Minor point:

Page 2: "This coupling is conserved among multiple flowering plant species" is an overstatement; so far clear evidence for coupling has been shown for *A. lyrata* and suggested for maize.

Response: We agree that this sentence over-stated the findings from our previous paper and we have adjusted the statement as suggested. The sentence now reads:

This coupling is conserved in *Arabidopsis lyrata* and may also operate in more distantly related maize¹³, suggesting it may play an important role in regulating DNA methylation homeostasis.

REVIEWERS' COMMENTS:

Reviewer #1 (Remarks to the Author):

I appreciate the efforts of the authors to revise the manuscript, and I found all my concern adequately addressed in the new version, as well as satisfying answers to the other reviewer's points. I still think that the redundancy in the title should be removed, and the argument "that many papers in the field use the term "epigenetic inheritance" to refer to inheritance over mitotic cell divisions" is not a good argument. Imprecise terminology applied by others should not be copied and propagated :-). But I can live with that!

Reviewer #2 (Remarks to the Author):

The author's made efforts to address my concerns. Regarding the novelty, I can follow their first argument; while I still think that the outcome of this work is not completely surprising, I agree that predictions should be tested. Nevertheless, I fail to see the novelty that heterochromatic regions respond differently to the loss of RdDM and ROS1 expression compared to euchromatic regions. RdDM has been shown to preferentially target small TEs located in euchromatic regions while the CMT2 pathway is mainly active at heterochromatic regions (Zemach et al., 2013). The authors cite this work, but do not discuss the obvious implications in connection to their work.

Response to comment 4:

I see the argument of the authors to present the data as boxplots. However, since no statistical test is included, a significance statement cannot be made (see line 115).

Response to reviewers' comments

Reviewer #1 (Remarks to the Author):

I appreciate the efforts of the authors to revise the manuscript, and I found all my concern adequately addressed in the new version, as well as satisfying answers to the other reviewer's points. I still think that the redundancy in the title should be removed, and the argument "that many papers in the field use the term "epigenetic inheritance" to refer to inheritance over mitotic cell divisions" is not a good argument. Imprecise terminology applied by others should not be copied and propagated :-). But I can live with that!

Response: We appreciate the reviewer's point of view, but have decided to retain "transgenerational" in the title.

Reviewer #2 (Remarks to the Author):

The author's made efforts to address my concerns. Regarding the novelty, I can follow their first argument; while I still think that the outcome of this work is not completely surprising, I agree that predictions should be tested. Nevertheless, I fail to see the novelty that heterochromatic regions respond differently to the loss of RdDM and ROS1 expression compared to euchromatic regions. RdDM has been shown to preferentially target small TEs located in euchromatic regions while the CMT2 pathway is mainly active at heterochromatic regions (Zemach et al., 2013). The authors cite this work, but do not discuss the obvious implications in connection to their work.

Response: We have added more to the discussion to address these comments, and further cite the work that demonstrates the preferential targeting of euchromatic regions by RdDM, and the high activity of the CMT2 pathway in heterochromatin.

Response to comment 4:

I see the argument of the authors to present the data as boxplots. However, since no statistical test is included, a significance statement cannot be made (see line 115).

Response: We have removed this use of the word significance to avoid confusion.